# Joint observations of oxygen atmospheric band emissions using OSIRIS and the MATS satellite

Björn Linder[1], Jörg Gumbel[1], Donal P. Murtagh[2], Linda Megner[1], Lukas Krasauskas[1], Doug Degenstein[3], Ole Martin Christensen[1], and Nickolay Ivchenko[4]

[1]Department of Meteorology, Stockholm University, Stockholm, Sweden
[2]Department of Space, Earth and Environment, Chalmers University of Technology, Gothenburg, Sweden
[3]Institute of Space and Atmospheric Studies, University of Saskatchewan, Saskatoon, Canada
[4]Division of Space and Plasma Physics, KTH Royal Institute of Technology, Stockholm, Sweden

**Correspondence:** Björn Linder (bjorn.linder@misu.su.se)

**Abstract.** The MATS (Mesosphere, Airglow/Aerosol, Tomography & Spectroscopy) satellite was launched in November 2022 and began collecting scientific measurements of the Mesosphere and Lower Thermosphere (MLT) in early 2023. The satellite utilises a multichannel limb-viewing instrument designed to gather images across six distinct spectral bands, each selected to capture atmospheric airglow from $O_2$ atmospheric band emissions as well as light scattered by noctilucent clouds (NLC). This article presents a comparison between the MATS limb measurements and the observations made by the OSIRIS spectrograph on the Odin satellite. Specifically, airglow signals from excited $O_2$, as recorded by MATS infrared (IR) channels and OSIRIS, are analysed over the polar regions under temporally and spatially aligned conditions. From December 2022 to February 2023, 36 close encounters of the two satellites were identified and analysed. The results show that the two instruments agree well on the overall structure but that the MATS signals generally exceed OSIRIS by $\sim 20\%$ in magnitude. OSIRIS measurements are also compared to the radiative transfer model SASKTRAN, to investigate stray light impact on the measurements.

## 1 Introduction

MATS (Mesospheric, Airglow/Aerosol, Tomography and Spectroscopy) is a Swedish satellite mission designed to investigate atmospheric dynamics in the Mesosphere and Lower Thermosphere (MLT) using tomographic and spectroscopic imaging (Gumbel et al., 2020). Since its launch in November 2022 from Te Māhia/Māhia Peninsula, New Zealand, the main instrument has taken over 4 million images of the MLT atmospheric limb, each separated into one of six spectral bands. Four channels target global atmospheric airglow emissions, whereas two target scattered light from Noctilucent Clouds (NLCs) observed in the MLT above the summer pole. The observed airglow emissions originate from the $O_2$ molecule in its $O_2(^1\Sigma_g^+ - ^3\Sigma_g^-)$ transition over a spectral region denoted the atmospheric A-band. As these images are taken through the atmospheric limb, both vertical and horizontal structures in the emitting layer can be observed. Using spectroscopy and tomography, the images are combined to derive three-dimensional temperature fields; fields that can be used to derive the properties of the gravity waves disturbing them (Linder et al., 2024). Measurements of $O_2$ atmospheric band airglow also offer insights into the composition

and chemistry of the MLT, as they enable the derivation of ozone and atomic oxygen concentrations. These key data products are also targeted by the MATS mission.

Odin was launched in February 2001, a joint atmospheric and astronomy mission, carrying the OSIRIS (Optical Spectrograph & Infrared Imager System) (Llewellyn et al., 2004). OSIRIS is a limb profiler that scans the atmosphere across UV, visible, and IR wavelengths at a spectral resolution of up to 1-2 nm. As the scanning height of the instrument reaches high altitudes, typically up to around 100 km, numerous scientific studies have been conducted targeting the MLT. These investigations include studying properties of NLCs (von Savigny et al., 2005; von Savigny and Burrows, 2007; Hultgren and Gumbel, 2014), retrieving mesospheric temperatures (Sheese et al., 2010), exploring interhemispheric coupling (Gumbel and Karlsson, 2011), and obtaining daytime ozone concentrations (Li et al., 2020).

This study identifies co-observations of the two satellites, in order to investigate how well the absolute calibrations of the two instruments agree. Calibration efforts of the limb imager have been substantial, both before launch and in orbit using stars (Megner et al., 2025). However, stray light effects have yet to be identified and characterised in the calibrated images (MATS L1b data product). Any such light, registered in the instrument but not directly connected to the observed scene, must be removed for MATS to correctly describe the temperature field of the MLT (the MATS L2 data product). An approach to evaluate the performance of the MATS limb imager is to consider the derived MLT temperature. The evaluation could then be performed by comparison with the temperature derived from ground-based instruments such as lidar, radar, and space-borne instruments such as SABER (Russell et al., 1999). However, performance evaluations are best conducted on lower-level data products to mitigate assumptions and discrepancies that may arise from the retrieval processes. As ground-based measurements of oxygen A-band emissions are not possible, spaceborne instruments are required, and preferably, comparison is made with another instrument that measures through the atmospheric limb. Two relevant missions can be considered. The first, MIGHTI on the ICON satellite (Englert et al., 2017) measured $O_2$ A-band and retrieved MLT temperatures (Stevens et al., 2022). The ICON operational period ended at the time of MATS launch, and this comparison would have to be climatological. Odin/OSIRIS allows for common-volume observations over the poles in the right spectral range, and the observational geometries are very similar to those of MATS. Unfortunately, OSIRIS is also known to be affected by stray light at higher scanning altitudes (Bourassa, 2003). SASKTRAN, a radiative transfer model designed for atmospheric research at the University of Saskatchewan, has a long history of application alongside OSIRIS measurements (Bourassa, 2003; Zawada et al., 2015). Using SASKTRAN, the contribution of stray light to OSIRIS measurements can be evaluated, and the absolute calibration of the instrument can be explored through comparisons with simulations of Rayleigh molecular scattering. Once OSIRIS has been characterised, the performance of MATS can be assessed.

In Section 2, the instrumentation of the satellites and their observation geometries are presented. Section 3 outlines the method for finding coincidences, and the processing to generate MATS-like data from OSIRIS is described. To enhance the understanding of stray light effects in OSIRIS data, a brief comparison is performed between the limb radiance profiles of SASKTRAN and OSIRIS. Sections 4 and 5 introduce the day- and nightglow comparisons between MATS and OSIRIS and the statistical evaluation is presented in Section 5.1. Finally, conclusions are given in Section 6.

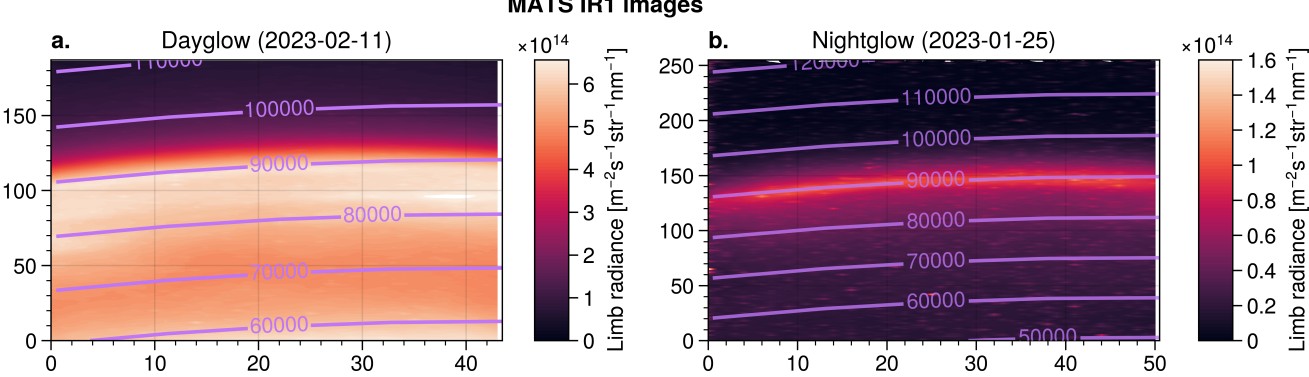

**Figure 1.** Two examples of IR measurements made by MATS in the studied period. The purple lines are tangent heights in meters. Note that the instrument settings have changed between the two dates, indicated by the different number of rows (horizontal axes) and columns (vertical axes) between the images, and the slight change in pointing of the instrument. The across-track extent is roughly 200 km at tangent point.

## 2 Background

In the MLT, emission in the visible and near-visible ranges is generated by multiple molecular species in different spectral regions. Many of these emissions originate from chemical reactions that occur during sunlit hours. Some processes are quick and generate emissions instantaneously; others lead to emissions that eventually occur at night. The emissions occurring during sunlit conditions are referred to as dayglow, whereas the emissions that occur in the shade of the Earth are referred to as nightglow. In the case of atmospheric band emissions from $O_2(^1\Sigma_g^+ - ^3\Sigma_g^-)$, the nightglow radiance is approximately an order of magnitude weaker than that of the dayglow emissions. Still, both day- and nightglow emissions are impossible to measure from the surface, as the lower atmosphere absorbs the radiation during its journey to the surface. For a satellite-based optical instrument, the absorption is considerably lower, especially when observing the emissions at higher MLT altitudes. However, in the lower range of the MLT, absorption is still significant and must be considered (Sheese et al., 2010). For evaluating satellite-borne measurements of A-band emissions, it is therefore convenient to compare instruments of similar observation geometry, so that absorption effects are similar.

### 2.1 Instrumentation of the MATS satellite

MATS operates in a dawn-dusk Sun-synchronous orbit at a local solar time of $\sim$ 17:30 at the ascending node, enabling global coverage at two different local times. The sun-synchronous orbit was a design choice allowing the solar panel to always face the sun and keep the instruments stored away from direct sunlight. The limb imager onboard the MATS satellite is equipped with a baffle that directs incoming light to a series of mirrors and beam splitters. The light from the observed scene is divided into six channels, each targeting a specific wavelength interval. Two main IR channels target the emissions made in the $O_2$ atmospheric

**Table 1.** The IR channels of the MATS limb imager.

| Channel | Central wavelength | Bandwidth | Target |
|---------|--------------------|-----------|--------|
| IR1 | 762 nm | 3.5 nm | $O_2$ A-band |
| IR2 | 763 nm | 8 nm | $O_2$ A-band |
| IR3 | 754 nm | 3 nm | Background |
| IR4 | 772 nm | 3 nm | Background |

band, while two IR channels target the Rayleigh-scattered background radiance in the spectral vicinity. The main channels
consist of a narrow spectral window (IR1) and a wider window (IR2). These windows are optimised so that the temperature
of the emitting gas can be derived, taking advantage of the wavelength distribution of the emitting $O_2$ gas that depends on
its temperature. The instrument further monitors two UV spectral ranges. These channels are named UV1 and UV2, and are
designed to target light scattered from noctilucent clouds. The treatment of these channels is outside the scope of this study.
Two example images taken by the IR1 channel are shown in Fig. 1 for nightglow (1a) and dayglow (1b). The images illustrate
how the dayglow field is intense and spread over a wide altitude range, whereas the nightglow emission typically is more
localised and weaker. The vertical range of the instrument is roughly 60-110 km, with a vertical sampling of approximately
250 metres at tangent height. The across-track field of view at the limb is approximately 200 km, with a horizontal sampling
of roughly 5 km, during nominal operations. The resolution is flexible in the sense that the binning (averaging) of columns and
rows in the image is controllable. The size of the image can also be adjusted, as seen by comparing the day- and nightglow
images in Fig. 1.

The observed light in the MLT is not solely from airglow emissions. In sunlit conditions, Rayleigh scattering is present at
these altitudes, and a portion of the light detected in the IR1 and IR2 channels comes from this background radiation. For
this purpose, the background channels IR3 and IR4, situated in the vicinity of the A-band emission spectral region, can be
used to subtract the contribution of the Rayleigh signal from the $O_2$ emissions. However, it should be noted that Rayleigh
scattered light is also subject to absorption by $O_2$, and therefore reduced in the main IR channels. Moreover, the stray light (i.e.
light registered in the instrument that originates outside the observed scene) entering the instrument is not accounted for by
background channel subtraction. During instrument design, suppression of stray light has been a priority. The stray light poses
a challenge for post-processing as its shape and magnitude may vary between channels. Comparing MATS images to those
from a comparable instrument can help to identify this light and characterise its behaviour. The IR channels of the MATS limb
imager are summarised in Table 1.

## 2.2 Odin and the OSIRIS instrument

Launched two decades ago, the Odin satellite has consistently provided the scientific community with data. Odin is in a Sun-
synchronous orbit at $\sim$ 19:00 at the ascending node. The OSIRIS spectrograph, one of the two main instruments, scans the
atmospheric limb from 30 to 110 km, obtaining radiance spectra between 280 and 810 nm at every tangent altitude. The A-

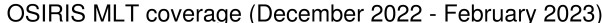

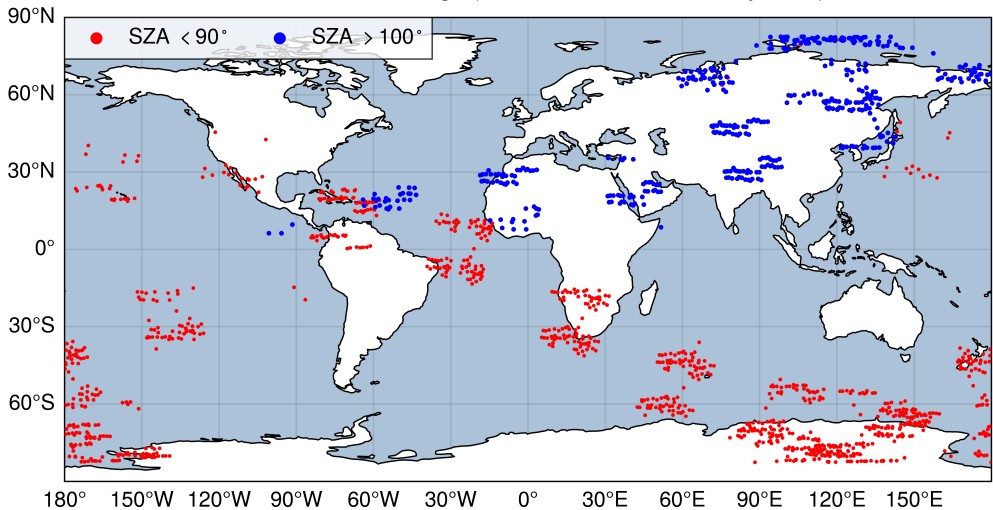

**Figure 2.** Odin MLT scan positions separated into dayglow and nightglow in the altitude range of 70-110 km from December 2022 to February 2023. Blue dots indicate nightglow conditions and red dots dayglow conditions.

band has specifically been used to retrieve the oxygen density of the mesosphere (Sheese et al., 2011) and the aforementioned mesospheric temperatures (Sheese et al., 2010). The full scan of the atmosphere takes approximately 2 minutes from the surface to the highest altitude, which in the studied period is typically around 100 km. More recently, Odin only occasionally made measurements of the MLT due to technical limitations, and a mere total of 2,145 MLT scans were identified in the period December 2022 - February 2023. Identified scans with well-defined night- and dayglow are illustrated in Figure 2, where

dayglow measurements are shown as red dots and nightglow measurements as blue dots. The distinction is based on the solar zenith angle (SZA) and is discussed in the next section.

## 3   Method

### 3.1   Isolating coincidences

To investigate nightglow emissions we consider measurements when the solar zenith angle (SZA) is greater than 100° at the

90 km tangent point. This ensures that there are no sunlit altitudes with significant scattering and/or emissions in the field of view. Similarly, dayglow measurements are compared under conditions in which all altitudes in the limb profiles *are* sunlit by ensuring that SZA < 90° at the tangent point. Due to the dawn-dusk orbits of the two satellites, during the solar equinox measurements are made above the solar terminator across all latitudes. In contrast, at the solstices, the satellites observe distinct sunlit and dark hemispheres, with clear dayglow and nightglow occurring in the opposite hemispheres. Ignoring November and

December, a period in which MATS instrument settings were still being optimised, Odin made 1194 MLT scans during January

through February, 579 scans fully in the dayglow regime, and 157 well-defined nightglow scans. Given that MATS images of the MLT were taken every 6 seconds, the number of conjunctions between the two instruments is limited by the fewer OSIRIS measurements. Additionally, the different local times of the satellites make close coincidences possible only near the poles. The timestamp used for the Odin measurements is the time the OSIRIS scan passes 90 km. As the spectrograph approximately scans the atmosphere at a rate of approximately 0.75 km/s, it takes 40 seconds to scan the altitude range 70 - 100 km, which corresponds to an along-track distance of 300 km.

### 3.1.1  Nightglow Selection

The comparison of nightglow measurements is non-trivial for several reasons: the signal-to-noise ratio is low for both instruments, there are significant spatial variations in the emitting field and highly variable auroral emissions are intense compared to the nightglow $O_2$ A-band emissions. Measurements made relatively close in space and time may thus deviate substantially in radiance. For these reasons, the nightglow conjunctions should have a strict selection criterion in space. Owing to the orbit geometries, this will also reduce the number of comparisons made near the auroral oval. Thus, a conjunction is defined as when the measurements are within 150 km and are separated at most by 30 minutes (see table 2). Only measurements made under well-defined nightglow are considered, i.e. SZA $> 100°$.

### 3.1.2  Dayglow Selection

Dayglow conjunctions are selected so that both OSIRIS and MATS perform their measurements under well-defined, sunlit conditions. Hence, measurements are only considered where the SZA at the 90 km tangent point is less than $90°$. The dayglow is considerably less spatially variable than the nightglow, and auroral emissions are less problematic, since they are not as prominent relative to the intense dayglow. Thus, slightly looser selection criteria could be used and dayglow conjunctions were defined as when the distance between the measurements is 250 km and 30 min (see Table 2). NLCs can create a strong and spatially variable signal in the IR channels that may cause the MATS and OSIRIS measurements to disagree even if they are close in time and space because their viewing orientations differ. However, since the presence of NLC is typically very clear, as it introduces strong vertical variations in an otherwise rather smooth limb profile and mainly affects a limited altitude, they are not that problematic.

### 3.2  Simulating MATS measurements using OSIRIS

OSIRIS MLT scans with a spectral resolution of 1-2 nm and can be translated to the MATS equivalent observations by spectrally integrating the OSIRIS spectra over the filter curves of the individual MATS channels. Figure 3 shows an example of a limb radiance spectrum from 750 to 775 nm, acquired from a tangent height of 85 km, as viewed by OSIRIS. The red bars illustrate the spectral range of the MATS IR filters. By integrating these spectral windows for every scanned altitude, limb radiance profiles corresponding to each MATS channel are obtained. These can then be compared with the actual measurements of the

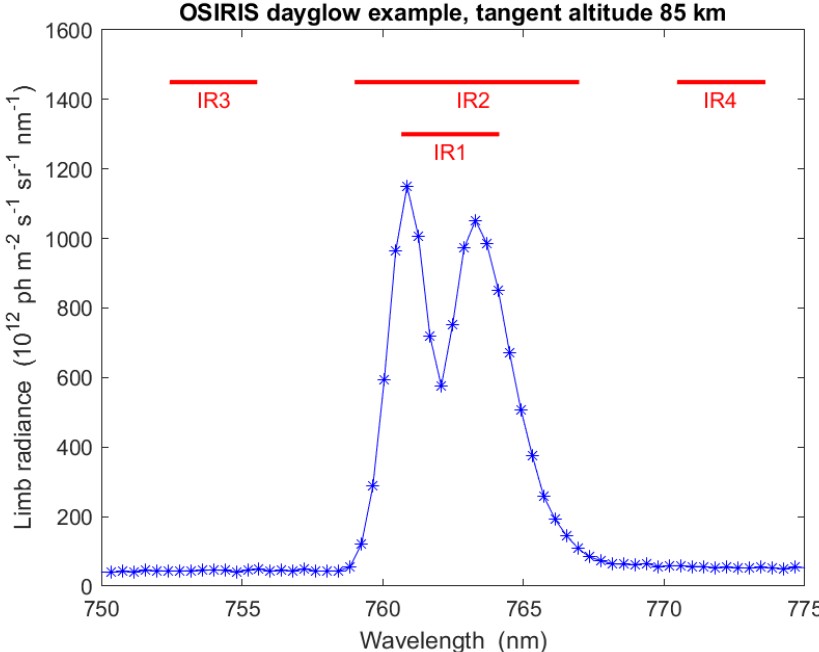

**Figure 3.** An OSIRIS spectrum obtained from a tangent altitude of 85 km. The spectral windows of the MATS IR channels are indicated by the red bars.

limb imager. As MATS images contain many of these profiles across its field of view, differences due to spatial variations can be explored by considering different image sections.

Figure 3 illustrates how IR1 and IR2 cover the atmospheric $O_2$ emissions, while the background channels IR3 and IR4 are located outside the emissions to quantify the Rayleigh-scattered and other background. For dayglow measurements, an estimate of the pure A-band emission signal can be obtained by removing a linear combination of the background channels from the IR1 and IR2 signals. However, this approach is not suitable for the L1b MATS data product as this has not been corrected for stray light effects, which affect the four IR channels to different extents. In the current paper, a comparison between the MATS and OSIRIS A-band signals is therefore performed without background subtraction. For each conjunction, we compare IR1 and IR2 separately with the OSIRIS equivalent. As the channels observe the same scene, the level of agreement between MATS and OSIRIS should be the same for both of these channels. If this is not the case, this is an indication of stray light effects or improper relative calibration.

### 3.3 Evaluating OSIRIS using SASKTRAN

To investigate the absolute calibration of the OSIRIS instrument, and to identify possible stray light effects, the OSIRIS data are compared to the SASKTRAN radiative transfer model (Bourassa, 2003; Zawada et al., 2015). SASKTRAN was developed to help in the retrieval of atmospheric constituents from OSIRIS, and in the model, the viewing geometry of the satellite can

be simulated to high precision. Simulating measurements involving A-band emissions is complex as a result of atmospheric chemistry, which requires extensive assumptions about the atmospheric state. To avoid this, limb scans are simulated in the spectral regions of the MATS background channels, IR3 and IR4, which only include weak airglow emissions and mainly observe Rayleigh scattered light (see Fig. 3). At lower altitudes, the relative contribution of stray light is limited, and a rough absolute calibration of OSIRIS can be derived. Higher up, deviations from SASKTRAN give an estimate of the stray light that perturbs the OSIRIS measurements. The assumption of surface albedo $\alpha$ should be included in the model, and since this is unknown from the circumstances of the measurement, simulations have been made for both a completely absorbing surface, $\alpha = 0$ and a perfect reflecting surface with $\alpha = 1$.

In Figure 4, the expected signal from the IR3 spectral window of SASKTRAN is compared with the corresponding OSIRIS observations. In the figure, the simulated and observed limb radiances, as well as the absolute difference between them, are presented for the twilight conditions with SZA $= 93° - 94°$ (4a and 4b) and daylight conditions with SZA $= 85° - 86°$ (4c and 4d). The results for the IR4 spectral window are similar (not shown).

For both twilight and daylight conditions, there is good agreement between OSIRIS and SASKTRAN in the intermediate range of 45 to 60 km (Figures 4a and 4c). This agreement is typically within a few percent, both during twilight and during daylight. In conditions with surface illumination, the level of agreement in the lower altitude range depends largely on the albedo used in the SASKTRAN simulation. Although the exact albedo conditions for the OSIRIS measurement are unknown, it is evident that the measured radiance is larger than that of the dark SASKTRAN case. The actual deviation is expected to be somewhere between the two extremes. At higher altitudes, the signal is dominated by stray light, and the OSIRIS measurements deviate substantially from SASKTRAN.

In daylight, the stray light signal decreases with the altitude of the tangent point and is approximately $3 \cdot 10^{13}$ m$^{-2}$ s$^{-1}$ str$^{-1}$ nm$^{-1}$ at 90 km in the IR3 spectral window (Figure 4d). This can be compared to typical dayglow signals measured by OSIRIS in the IR1 and IR2 bands, which are typically an order of magnitude higher ($10^{14}$ m$^{-2}$ s$^{-1}$ str$^{-1}$ nm$^{-1}$, see Section 4). Thus, stray light contribution from Rayleigh scattering should not substantially affect the OSIRIS dayglow measurements, especially in the IR1 and IR2 spectral windows, where a large portion of light originating from lower altitudes is absorbed in the A-band on its way upwards.

Concerning stray light originating from the airglow emissions in the $O_2$ A-band itself, the SASKTRAN simulations provide little information. Although it is not possible to quantify the intensity of these disturbances, it is feasible to assess their vertical structure. Figures 4a and 4b demonstrate that stray light effects are relevant even during twilight conditions when the Sun has set at lower atmospheric levels. The light originating from high, sunlit regions independently produces an unintended signal within the instrument. Panel 4b further shows that this stray light can be expected to be fairly constant throughout the OSIRIS scan between 70 km and 100 km. This is important for the OSIRIS / MATS comparisons in Section 5.1, since OSIRIS measures nightglow emissions under similar conditions, when light originates only from the MLT.

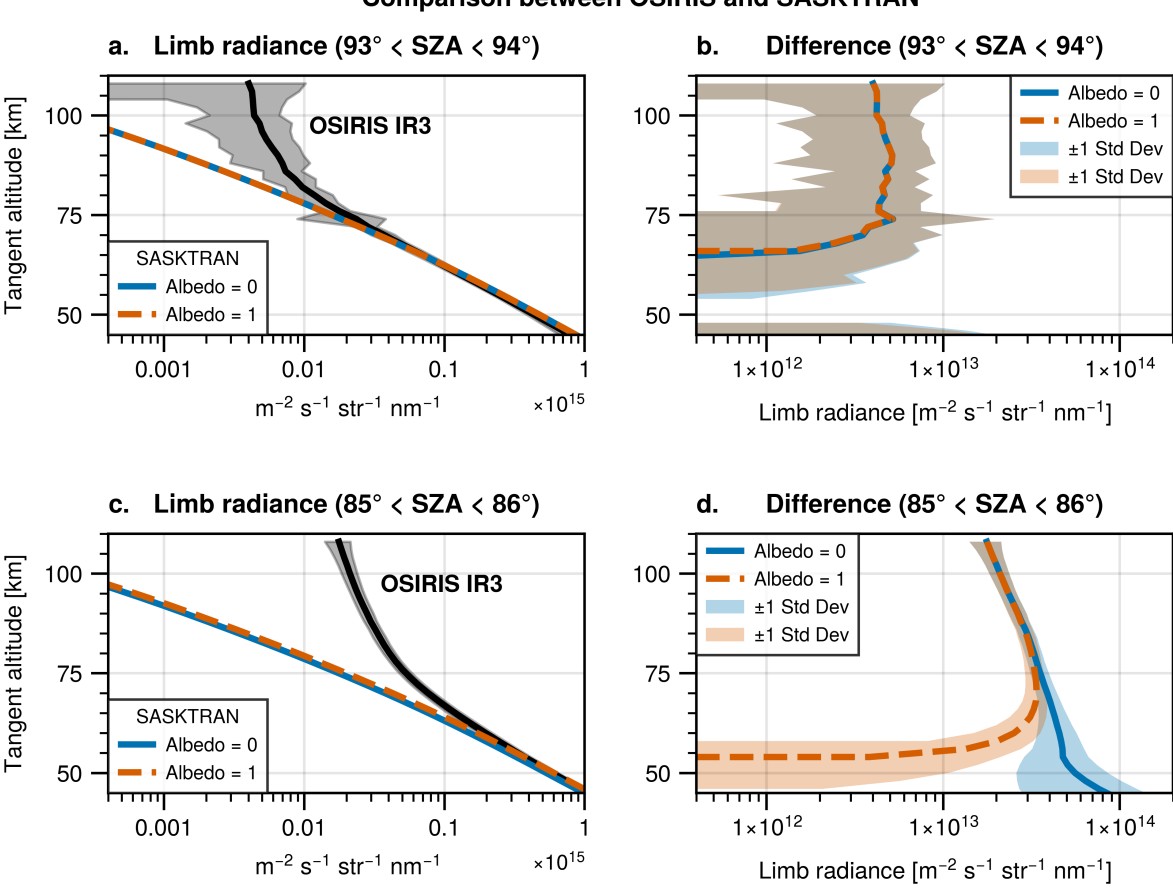

**Figure 4.** OSIRIS over the IR3 spectral window compared to IR3 simulated using SASKTRAN. Panel a) and c) illustrate the limb radiances for SZA = $93°$ - $94°$ and $85°$ - $86°$, respectively. Panels b) and d) show the absolute difference in limb radiance for the same SZA. Two different surface albedos are used in the simulations, indicated by the two different coloured lines and corresponding shading.

**Table 2.** Selection criteria and number of MATS / Odin conjunctions identified during January and February 2023. Two nightglow conjunctions from December 2022 are also included.

|  | Dayglow (SZA $< 90°$) | Nightglow (SZA $> 100°$) |
|---|---|---|
| Max distance | 250 km | 150 km |
| Max time difference | 30 min | 30 min |
| No. of coincidences | 20 | 16 |
| No. MATS images | 1133 | 420 |
| No. days | 6 | 7 |

A coincidence is determined to occur when the separation of the 90 km tangent point locations is smaller than the max distance and the difference in measurement time is within the max time difference.

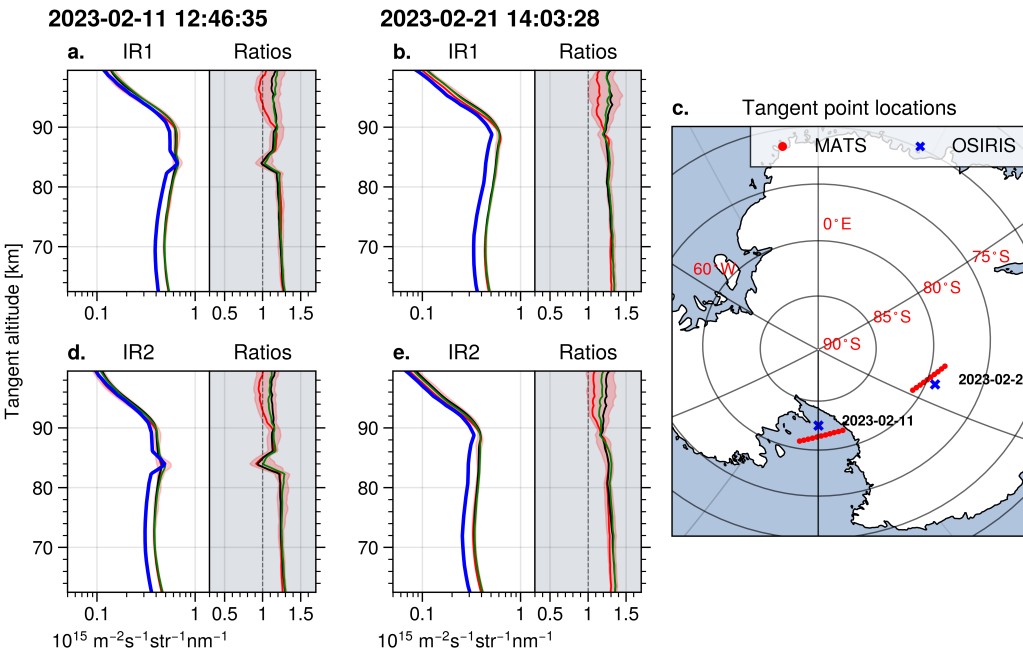

**Figure 5.** Dayglow conjunctions obtained during February. One example is from 11 February 2023 and one from 21 February 2023. The blue lines are limb scans by OSIRIS. Red, black, and green lines depict limb radiance in the left, middle, and right sections of the MATS limb imager. The grey inlets illustrate the ratio between the two instruments. The shaded red region illustrates the span of the values.

## 4 Dayglow Examples

20 dayglow coincidences between MATS and Odin were identified in February 2023. Due to the high sampling rate of MATS, approximately 25-65 MATS images are taken within the selection criteria for each OSIRIS measurement, totalling 1133 MATS

dayglow images. The coincidences are located over the south pole, where sunlit conditions prevail during the studied period. All coincidence profiles identified during dayglow conditions are presented in Appendix A1. Typically, the limb radiance profiles obtained from MATS in the temporal and spatial vicinity of the OSIRIS measurement do not show significant variability, especially not below the tangent altitude of the emission peak. Two example coincidences are shown in Figure 5, from 11 February 2023 and 21 February 2023. The blue lines represent OSIRIS measurements, shown alongside red, black, and green lines that depict the limb radiances of the left, middle, and right portions of the MATS images. The red shaded area highlights the span between the minimum and maximum signals detected around the OSIRIS measurement. On 11 February 2023, there is an NLC in both the OSIRIS and MATS measurements, which is seen as a peak at around 84 km. On 21 February 2023 the measurements of the two satellites were made 6 minutes apart, and on 11 February 2023 the measurements were made 15 minutes apart. For dayglow comparisons, the most striking is that there is very little variability across all MATS measurements corresponding to a given OSIRIS measurement. The red shading, which marks the area between the maximum and minimum, is hardly visible below 90 km. The red, black, and green lines that illustrate the different sections of the MATS images are also difficult to distinguish. From the ratios, it is clear that for all altitudes, MATS and OSIRIS seem to differ by approximately 20% - making MATS consistently larger than the corresponding OSIRIS measurement. At higher altitudes, where emissions are reduced, variability increases, but the general agreement is consistent with the 20% difference found below 90 km. For the statistical analysis in Section 5.1 , the conjunctions substantially influenced by NLC were excluded (this was considered to be one, namely 11 February 2023 in Figure 5), due to the strong local effect on the signal, which, in contrast to the airglow signal, will differ even for minor differences in viewing conditions.

## 5  Nightglow Examples

Due to stricter nightglow coincidence criteria and several coincidences perturbed by auroras, the nightglow analysis was extended to include the month of December, even though the data from December are sporadic. A total of 16 nightglow coincidences between MATS and Odin were identified from December 2022 through February 2023. The time of year of the available data means that, in contrast to the dayglow measurements, the nightglow measurements are located around the north pole. The comparisons of the two instruments during nightglow measurements are instructive due to the reduced impact of Rayleigh scattered stray light. However, the signal levels are strongly reduced as compared to dayglow, rendering the measurements more vulnerable to minor stray light influences, residual dark current in the instrument, and post-processing (e.g. desmearing and nonlinear corrections) that may leave residual signal.

At high altitudes, above the expected extent of the limb radiance of the nightglow layer, it is anticipated that the radiance would diminish to zero without auroral activity. Nonetheless, MATS images generally show a residual signal at the top of the image. This signal may be due to stray light influence or imperfection of the post-processing corrections (Megner et al., 2025). To properly compare the weak nightglow signals between MATS and OSIRIS, this signal needs to be removed. Assuming the residual signal to be homogeneous throughout the image, it can be removed by simply subtracting the signal of the top rows (above 110 km) of the image of every nightglow measurement from the entire image.

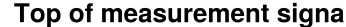

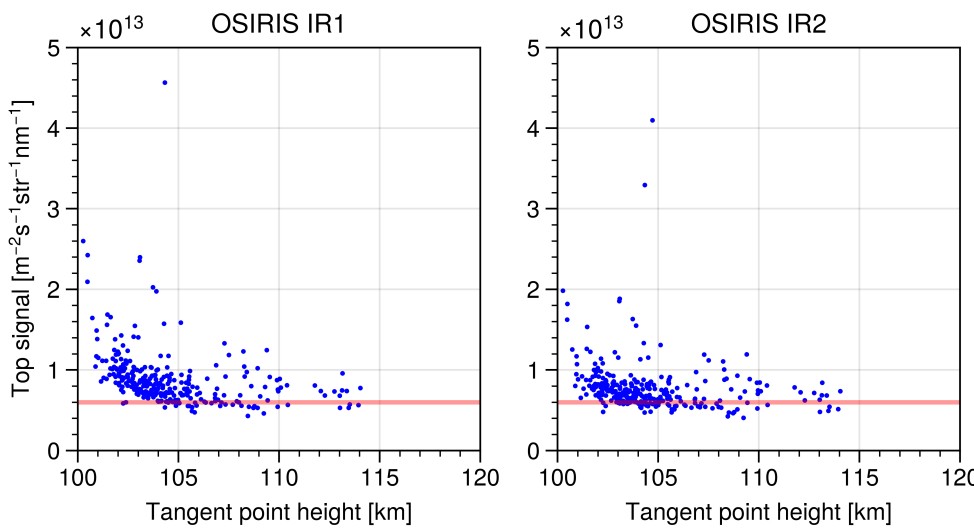

**Figure 6.** The signal at the top of OSIRIS measurements during nightglow conditions as a function of tangent height. The nightglow signal is reduced with increasing tangent height, but an offset remains.

In the period studied, OSIRIS mainly scans the atmosphere up to a tangent altitude of $\sim 105$ km and rarely above. Hence, as opposed to MATS, identifying potential remnants of dark current and stray light effects from the emission layer below becomes a more challenging task for OSIRIS as the signal at the top of the measurement may contain $O_2$ nightglow. Figure 6 illustrates the signal at the top of the OSIRIS nightglow measurements as a function of the tangent altitude of the top. The figure clearly illustrates that the signal decreases with altitude in both IR1 and IR2 simulations, most significantly up to 103 km. At higher altitudes, the signal stabilises at a nearly constant value. The value is similar for the two channels, approximately $0.65 \cdot 10^{13}$ m$^{-2}$ s$^{-1}$ str$^{-1}$ nm$^{-1}$. These values were subtracted from the OSIRIS data before the comparison to the measurements made by MATS, assuming their origin to be stray light from the emission field below and/or dark current remnants.

All of the obtained nightglow coincidences are presented in Appendix A2. The 16 coincidences that span 420 MATS images within the selection criteria illustrate how variable the nightglow field is. Two illustrative examples, from different nightglow conditions, are shown in Figure 7. Figures 7a and 7d were made on 11 January 2023 when the emissions were weaker and more variable. The large red area marks the radiance span between the minimum and maximum of all measurements and illustrates the variability along the orbit. The variability is also shown by the differences between the red, black, and green profiles, which again indicate variations across the MATS images. Figures 7b and 7e are examples of measurements made during stronger nightglow emissions, where variations in time and space are more constrained. In this example, the variations along the orbit and across the images are small. The ratios again indicate that MATS measures $\sim 20\%$ higher limb radiance than OSIRIS. In the following statistical analysis, nightglow conjunctions influenced by auroral activity, characterised by significant fluctuations at either low or high tangent heights or captured near times of observed auroral activity in other images, were omitted. The

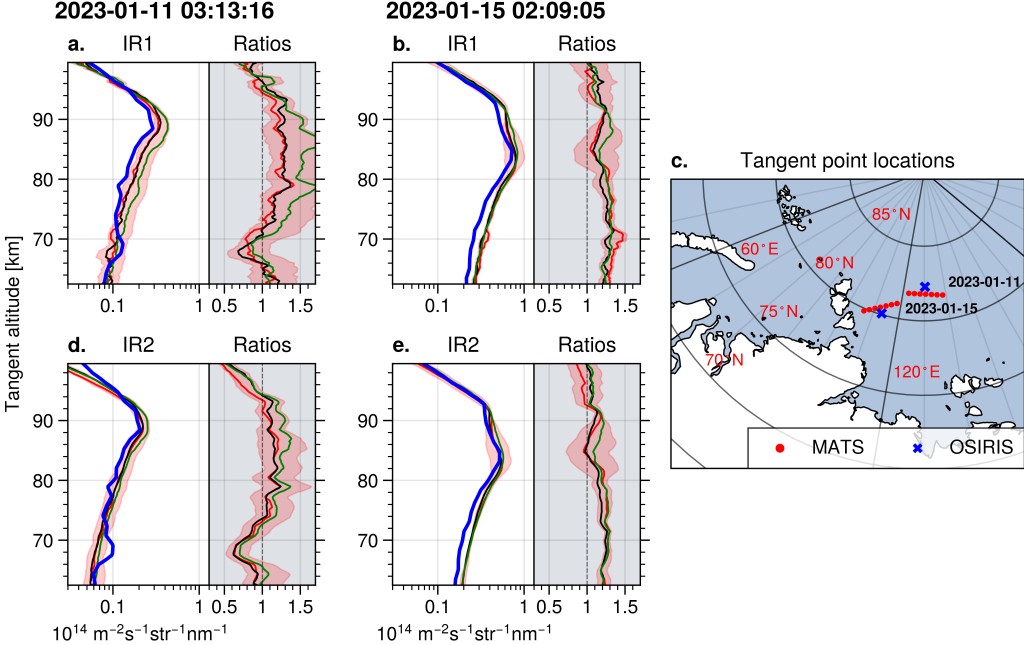

**Figure 7.** Examples of nightglow conjunctions obtained during January. The left column is from a conjunction on 11 January 2023 while the right column is from 15 January 2023. Blue lines illustrate limb radiance measured by OSIRIS. Limb radiances as observed in the left, middle, and right portions of the MATS images are depicted by red, black, and green lines, respectively. Red shading highlights the range between observed minima and maxima. Inlets display the signal ratios.

ignored conjunctions consist of the first pass on the 24th of December, all the passes on the 3rd, the second on the 9th of January, and the second on the 15th of January. Most of these indicate auroral perturbations. This reduces the total number of nightglow coincidences to 8 spread across 193 MATS images.

## 5.1 Statistical evaluation and discussion

As indicated by the examples in the previous sections, the comparisons between MATS and OSIRIS yield similar results during dark and sunlit conditions. Based on all coincidences, the mean difference between OSIRIS and MATS is shown in Figure 8, for the day- and nightglow separately. The blue and red lines display the mean differences for IR1 and IR2, subtracting OSIRIS from the MATS measurements, with the error of the mean indicated by the dark-shaded areas. The variability (one standard deviation) of the different observations is shown as light-shaded areas. The inlaid panels show the relative difference for both channels. Relative differences were calculated as (MATS - OSIRIS) / OSIRIS.

The nightglow comparison suggests that the radiances reported by the MATS IR1 channel appear to be approximately $20\%$ above the OSIRIS measurements for altitudes between 80 and 90 km. The relative agreement is better at higher altitudes. Smaller-scale variations also occur, likely because of the emission field's large variability and the relatively small sample size.

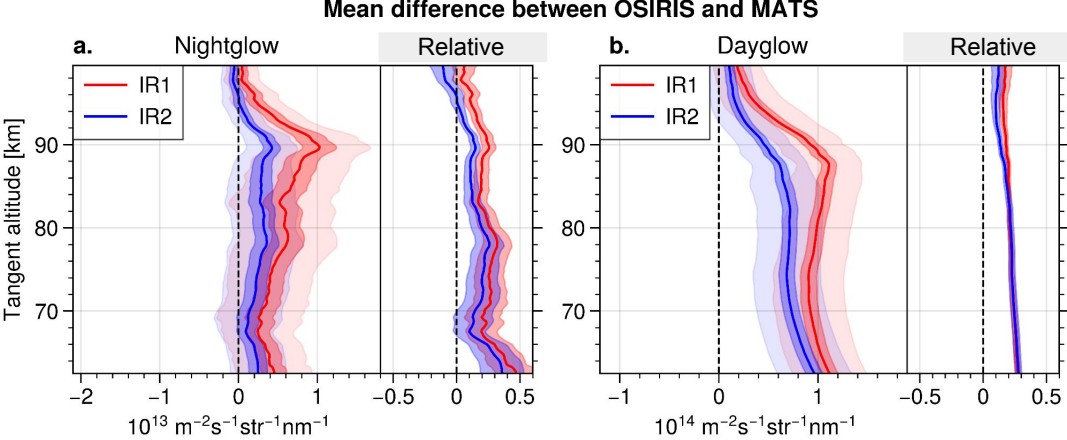

**Figure 8.** The mean of the difference of the limb radiance, subtracting OSIRIS from the MATS measurements for the two channels IR1 (red) and IR2 (blue), for nightglow (SZA > 100°, panel a.) and dayglow (SZA < 90°, panel b.). The solid lines show the mean of the differences with the error of the mean indicated by the dark-shaded areas, and the one standard deviation variability marked by the light-shaded areas. The right-hand sub-panels show the relative differences similarly, but the one standard deviation variability has been omitted for clarity reasons.

There is a small difference in how well IR1 and IR2 agree with OSIRIS. The ratios are very sensitive to instrument offsets. For example, if OSIRIS measured $0.1 \cdot 10^{13}$ m$^{-2}$ s$^{-1}$ str$^{-1}$ nm$^{-1}$ less in the IR2 spectral window, the relative difference would be the same for the two channels.

Under dayglow conditions, the situation is similar but less variable, the latter likely due to the larger signal and sample size. As illustrated in Figure 8b, the agreement between OSIRIS and MATS increases with altitude, although not as pronounced as for nightglow conditions. MATS consistently measures higher limb radiance than OSIRIS, and the relative difference is $\sim 20\%$ for both IR1 and IR2. The daytime measurements of MATS IR1 and IR2 are known to be affected by baffle scattering at the upper edge, and some level of stray light can be expected throughout the image. However, subtracting a constant signal from the entire image, like for nightglow, only reduces the relative difference by a few percent in the lower altitude range, with the effect tapering off to zero above 90 km.

From the discussions in Section 3.3, it is clear that for OSIRIS, stray light from molecular Rayleigh scattering does not dominate the dayglow measurements. Stray light originating from the A-band emissions at high altitudes could have a substantial contribution, but based on the SASKTRAN simulations, the signal should be fairly constant across the profiles. Such an offset would lead to a proportionally larger signal at higher altitudes and could explain the better agreement at higher altitudes.

# 6 Conclusions

We have compared MATS and OSIRIS measurements during both day- and nightglow conditions. Under both conditions, the differences between the instruments are similar. MATS measures approximately 20% stronger limb radiance in the altitude range of 70-90 km than OSIRIS. The two instruments agree better at higher altitudes than at low altitudes. The dayglow measurements show a more consistent result between IR1 and IR2, in that both channels show the same relative difference, and that it remains rather constant with altitude. In nightglow, there is more variability with altitude and there is an indication of a

minor difference between the two channels with IR1 observing a $20 - 25\%$ higher intensity than OSIRIS and IR2 a $15 - 20\%$ higher intensity at the peak of the airglow layer. However, these minor differences between the channels are on the boundary of statistical significance and could therefore be explained by the limited nightglow sample size. If the assumption about no stray light or uniform stray light in MATS nightglow measurements holds, the difference between the instruments is solely due to differences in absolute calibration.

Stray light constitutes a limitation of this comparison study. Based on simulations in SASKTRAN, stray light is expected to be observed during both day- and nightglow conditions in OSIRIS measurements, but the exact amount of stray light from the $O_2$ A-band emissions is unknown. In addition, the MATS measurements can be expected to be susceptible to stray light. The data suggest the presence of stray light in the upper part of the MATS images, which may provide an explanation for the higher signal levels of MATS compared to OSIRIS.

The absolute calibration of the OSIRIS measurements has never been officially validated. Unpublished investigation at the beginning of the mission focused on limb-scattered sunlight at lower altitudes. This is particularly important for the A-band emissions targeted in this study, since dim, high-altitude emissions require more precise knowledge of parameters like dark current and detector nonlinearity. The absolute error of the MATS calibration for the channels in question is reported as 3-4%, excluding stray light effects (Megner et al., 2025). The satellite was designed to have an absolute error within 10%,

in accordance with mission requirements. Given the unknown error in the high-altitude OSIRIS calibration, an agreement within 20% between the two satellite instruments is quite acceptable. Stray light effects in MATS images could provide an explanation for some of this difference. The quantification and characterisation of the MATS stray light is ongoing work. This will be particularly important for the temperature retrieval from the $O_2$ A-band during the day, where the ratio of signals obtained from IR1 and IR2 must be used.

*Code availability.* Software code is available on request.

*Data availability.* The OSIRIS data is available from the University of Saskatchewan at https://research-groups.usask.ca/osiris/data-products.php. The MATS level 1b dataset can be accessed from the Bolin centre database at https://doi.org/10.17043/mats-level-1b-limb-cropd-1.0. The code used to produce the level 1a and level 1b datasets is available on github repositories https://github.com/innosat-mats/level1a and https://github.com/innosat-mats/MATS-L1-processing, respectively.

*Code and data availability.*

*Sample availability.* TEXT

*Video supplement.* TEXT

## Appendix A: All conjunctions

### A1 Dayglow coincidences

### A2 Nightglow coincidences

*Author contributions.* BL: formal analysis, writing, visualisation, JG: processing of OSIRIS data, supervision, LM: supervision, writing, DM: SASKTRAN simulations, DD, OMC, LK & NI: conceptualization and discussions

*Competing interests.* JG is a member of the editorial board of Atmospheric Measurement Techniques.

*Disclaimer.* TEXT

*Acknowledgements.* B. Linder, D. P. Murtagh, O. M. Christensen, L. Krasauskas, L. Megner, and J. Gumbel received funding from the Swedish National Space Agency (grant nos. 2012-01684, 210/19, 2021-00052, and 2022-00108). The authors acknowledge the work of Jacek Stegman, Jonas Hedin, and Joachim Dillner, who are also involved in the MATS satellite mission, and Aglaja Roth for her contribution to the evaluation of the instrument.

**Dayglow IR1**

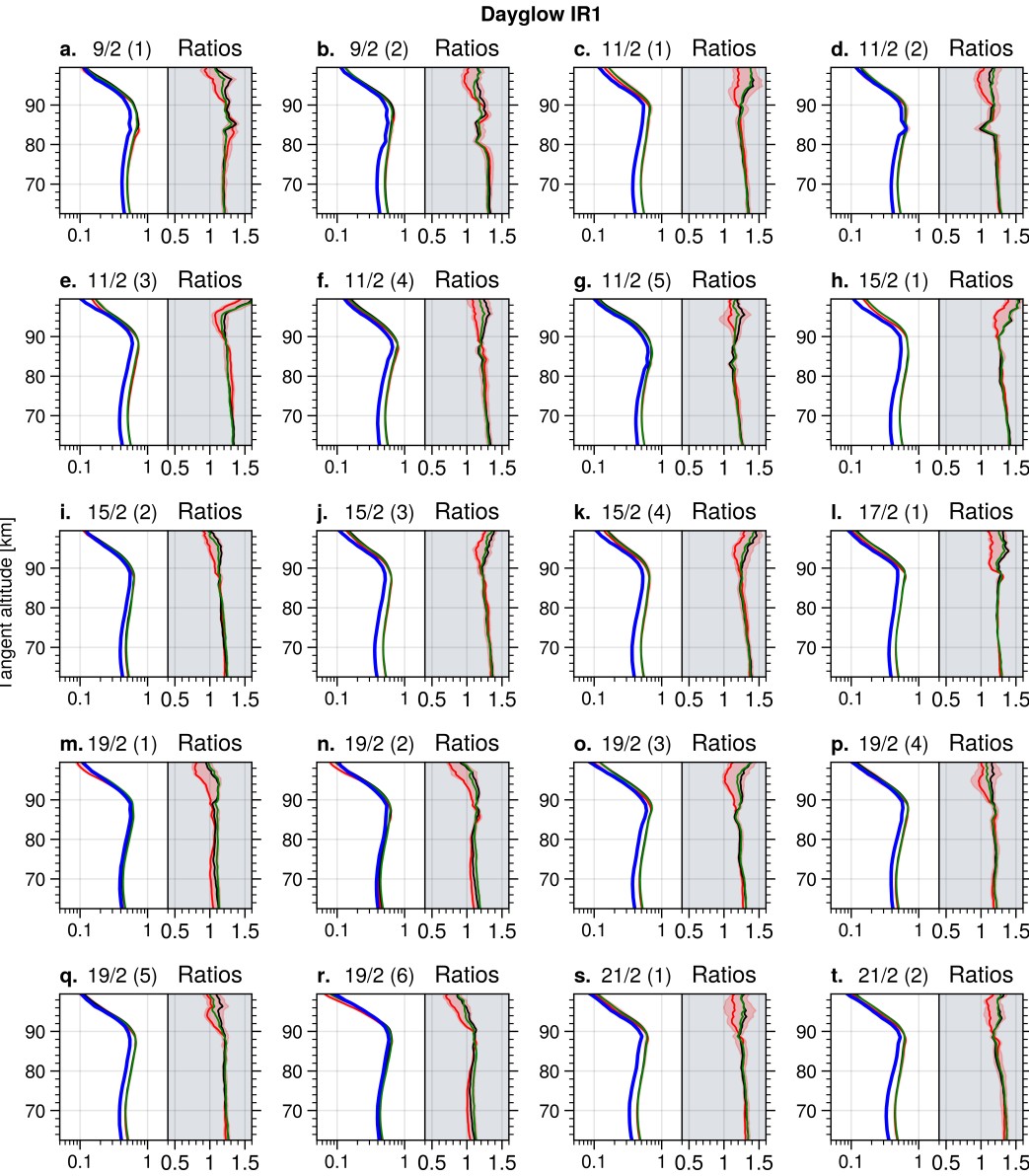

**Figure A1.** IR1 comparisons for SZA < 90°. The limb radiance is in units of $10^{15}$ m$^{-2}$ s$^{-1}$ str$^{-1}$ nm$^{-1}$.

**Dayglow IR2**

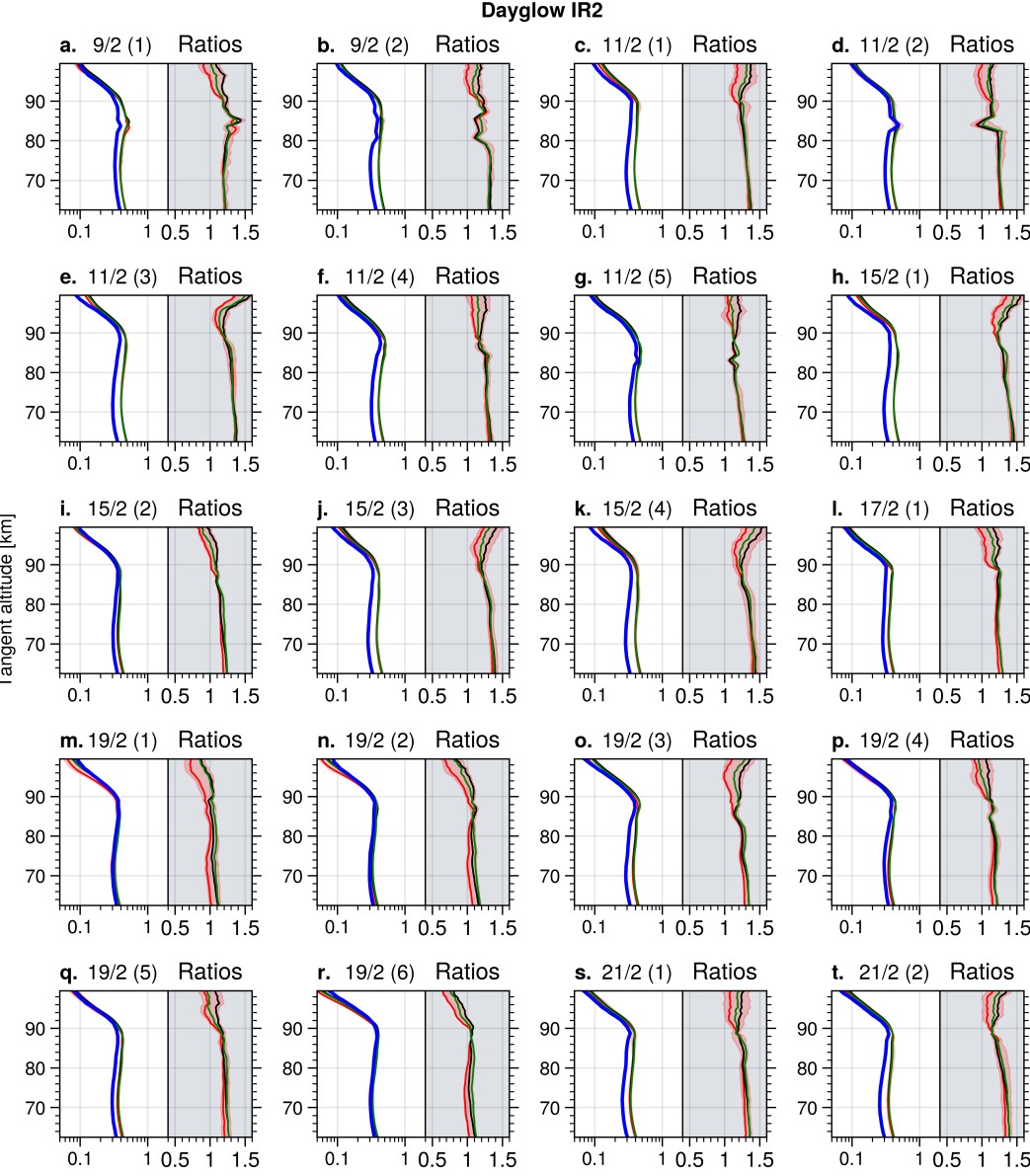

**Figure A2.** IR2 comparisons when SZA < $90°$. The limb radiance is given in $10^{15}$ m$^{-2}$ s$^{-1}$ str$^{-1}$ nm$^{-1}$.

**Nightglow IR1**

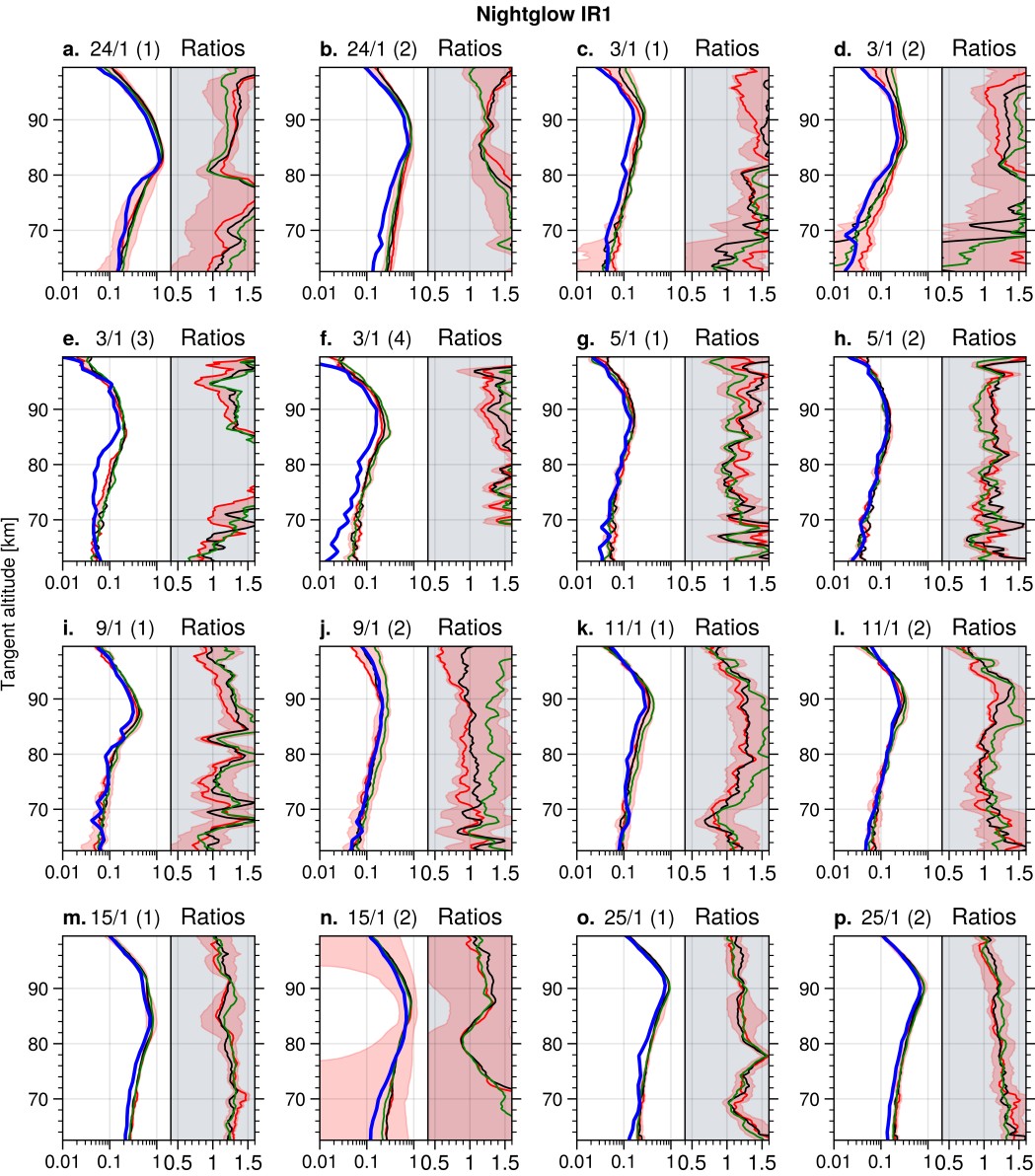

**Figure A3.** IR1 comparisons when SZA > $100°$. Limb radiance is given in units of $10^{14}$ m$^{-2}$ s$^{-1}$ str$^{-1}$ nm$^{-1}$.

**Nightglow IR2**

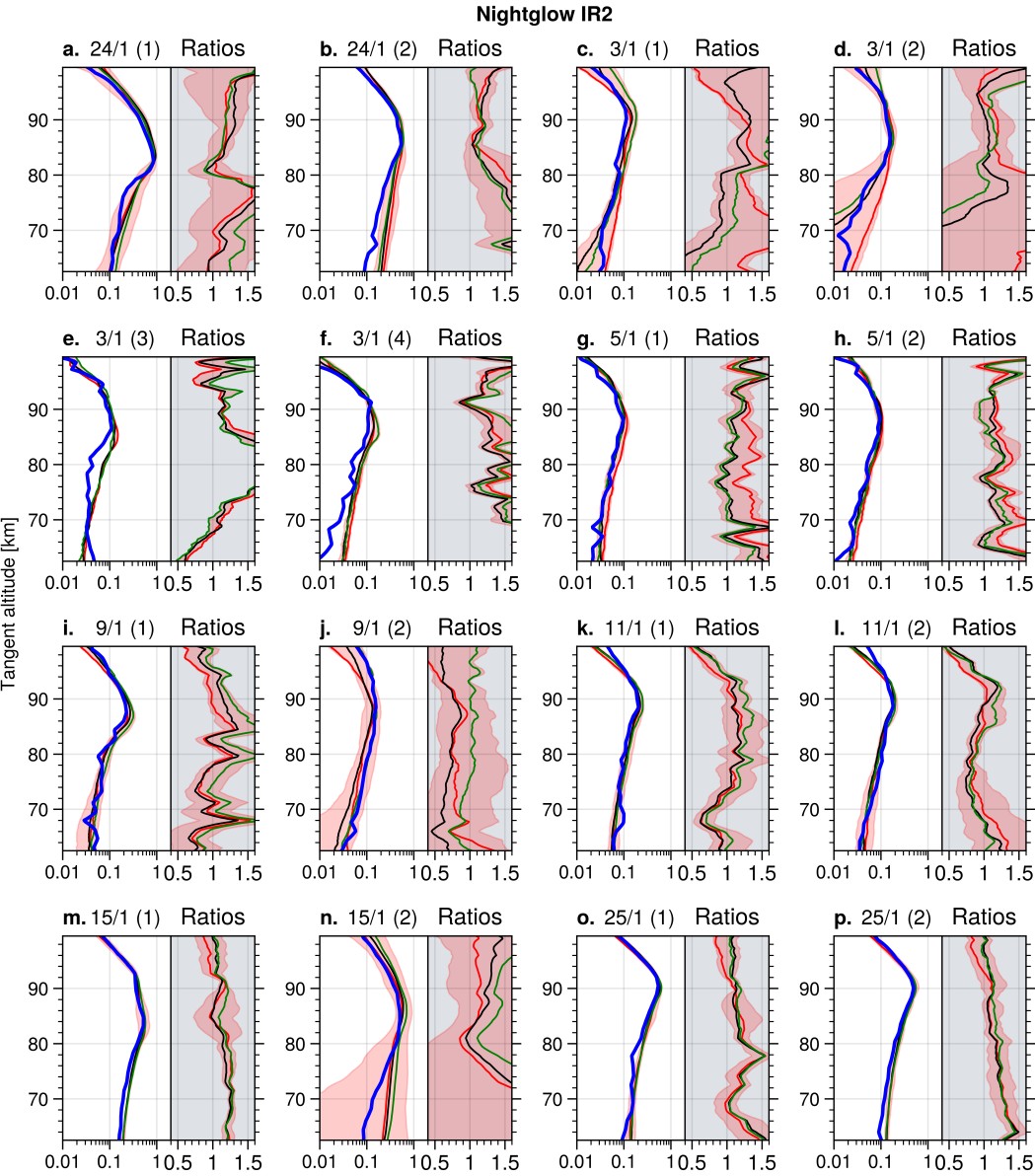

**Figure A4.** IR2 comparisons when SZA > $100°$. The limb radiance is in units of $10^{14}$ m$^{-2}$ s$^{-1}$ str$^{-1}$ nm$^{-1}$.

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
