# Peer review of "Joint observations of oxygen atmospheric band emissions using OSIRIS and the MATS satellite"

_EGUsphere, 2025_

## Author Response (AR1)

**Author response to Referee #1**

We would like to thank Referee #1 for the valuable comments. We have included the comments one by one, in bold text, along with our answers. If lines are given in the answers they refer to the original manuscript. The blue colour indicates text added in the revised manuscript.

**General comments:**

**The manuscript describes an evaluation of limb radiance measurements of the $O_2$ atmospheric band near 760 nm taken with the multi-channel imager of the MATS satellite launched in November 2022. The data are compared with spectra of the limb profiler OSIRIS on the Odin satellite (launched in February 2001) in the same wavelength range. Only a small sample of 20 dayglow and 16 nightglow coincidences with sufficient agreement in time and location could be selected. On average, the $O_2$ limb radiance is about 20% higher than in the case of OSIRIS. It is assumed that differences in the absolute calibration are the most likely explanation for the discrepancies. However, the impact of stray light remains uncertain for both instruments, even with radiative transfer calculations with SASKTRAN.**

**For research in the Earth's mesosphere and lower thermosphere, MATS is a promising mission. Therefore, I appreciate the submission of the manuscript by Linder et al. to AMT as it allows insights in the performance of MATS. Nevertheless, I would like to see some improvements before the start of the publication process. In particular, the design of the investigation could be better motivated. Some clarifications might be provided by another MATS-related manuscript submitted by Megner et al.. However, the present manuscript should be sufficiently self-explanatory.**

**Specific comments:**

☐ **Sect.1: The introduction only describes MATS and OSIRIS in a rather technical way. For a better understanding of the goals of these airglow-observing instruments and the importance of tackling the calibration issues, it would be helpful to discuss them in a broader context. There is no information on other satellite missions (or maybe ground-based observations, although the A-band cannot directly be accessed) that could contribute to solve the calibration issues. As MATS and OSIRIS data appear to have calibration and stray light**

**issues, it would be important to understand why the study was designed in the described way.**

We agree that the manuscript did miss some motivation as to why the evaluation was designed as it was. We have now included a part motivating the design of the study in Section 1. We have also added a short note on atmospheric band airglow in the introduction.

☐ **Fig.1: The rough extent of the images in horizontal direction could also be provided in the caption and not only in the text where it is more difficult to find.**

This is a good suggestion. We have now updated the figure caption to include the across-track extent.

Two examples of IR measurements made by MATS in the studied period. The purple lines are tangent heights in meters. Note that the instrument settings have changed between the two dates, indicated by the different number of rows (horizontal axes) and columns (vertical axes) between the images, and the slight change in pointing of the instrument. The across-track extent is roughly 200 km at tangent point.

☐ **L.147: "MATS background channels, ..., which do not include airglow emissions": This statement is not fully correct as there are faint emissions of other $O_2$ bands, bands of OH, and airglow (pseudo-)continua. Nevertheless, the related measurement errors for the A-band should be very small.**

This is correct. The manuscript has been updated accordingly:

L146: To avoid this, limb scans are simulated in the spectral regions of the MATS background channels, IR3 and IR4, which only include weak airglow emissions and mainly observe Rayleigh scattered light (see Fig. 3).

☐ **Fig.5: The red, orange, and green lines are hard to distinguish. A remark in the caption could be helpful here. Moreover, the orange lines are difficult to see on a orange/pink background ("red shaded area" in the text), which is not explained in the caption. Would it be possible to use a different line colour or to change the colour of the shaded areas?**

Fig. 5 has been updated and the orange line is now black, which stands out more. References to the orange line in text have been updated. The red shaded area is now explained in the caption. The figures in the appendix have been updated to have the same colours.

☐ **Fig.7: Similar to Fig. 5, it is difficult to recognise the individual lines. Again, the orange lines are most challenging.**

Fig. 7 has been updated with new colours and the orange line is now black. The text has been changed accordingly. The lines of the figures in the appendix now have the same colours.

☐ **Sect.6: As the manuscript appears to describe the first study that actually evaluates real MATS data, it would be interesting know more about the performance of the satellite/instrument compared to the expectations before the launch. At least for the calibration of the airglow radiance and stray light issues (i.e. the topic of the study), more information would be helpful. In this context: where does the statement "is reported as 3-4%" come from?**

The calibration paper is at the time of writing also under review (finalising response). While this is one of the two first studies that evaluates MATS data, these questions are treated in greater detail in the calibration paper. The 3-4% is from this calibration paper (Megner et al.), which is now cited alongside the statement. From mission requirements, an absolute error within 10% was required - this is included in the updated manuscript.

As for the expectation with regards to stray light, in short, the suppression of stray light was prioritised during the development of the limb imager and the presence of any such light is unfortunate for the mission. Nonetheless, it was anticipated that some stray light would be detected in the measurements and its characterisation was scheduled already prior to the mission launch. The exact magnitude of the stray light and its behaviour is the focus of ongoing work.

**Technical corrections:**

☐ **Inconsistent spelling: "stray light" and "straylight" are used.**

We have replaced "straylight" with "stray light" throughout the manuscript.

☐ **Fig.8: In "(SZA > 100°), panel a)", the parentheses are not consistent with "(SZA < 90°, panel b)".**

Corrected.

**Author response to Jaroslav Chum**

We would like to thank Jaroslav Chum for the valuable comments. We have included the comments one by one, in bold text, along with our answers. If lines are given in the answers they refer to the original manuscript. The blue colour indicates text added in the revised manuscript.

**General comment**

The manuscript compares the intensities of atmospheric band of $O_2$ air glow emission measured simultaneously by OSIRIS and MATS satellite. I believe that independent simultaneous measurements of air glow intensities are useful and interesting for community analyzing and interpreting the airglow data and I consider the topic suitable for Atmospheric Measurement Techniques. There are, however, several points that should be addressed more carefully before the publication. See the specific comments.

**Specific comments**

☐ **Introduction, line 24, "As the scanning height of the instrument reaches high altitudes…" Specify the heights.**

The altitude, typically around 100 km, has been added:

L23: OSIRIS is a limb profiler that scans the atmosphere across UV, visible, and IR wavelengths at a spectral resolution of up to 1-2 nm. As the scanning height of the instrument reaches high altitudes, typically up to around 100 km, numerous scientific studies have been conducted targeting the MLT.

☐ **Section 2.1, line 57, "MATS operates in a dawn-dusk Sun-synchronous orbit…" The dawn-dusk sector is rather atypical for airglow measurements. It would be useful to discuss/explain the reason for selecting such an orbit.**

The orbit was selected as it was beneficial for the construction of the satellite, as the solar panel can then be fitted on a side of the satellite that will face the sun. Further, the instruments are then hidden away from direct sunlight behind the panel, which reduces straylight. A note on this is added in the updated manuscript.

L57: MATS operates in a dawn-dusk Sun-synchronous orbit at a local solar time of ˜ 17:30 at the ascending node, enabling global coverage at two different local times. The sun-synchronous orbit was a design choice allowing the solar panel to always face the sun and keep the instruments stored away from direct sunlight.

☐ **Section 2.1, It would be good to specify the spectral ranges of all IR filters/channels here in this Section. A short note about the remaining channels (5, 6) would also be useful.**

We think that this is a good suggestion. A small table has been added to Section 2.1:

IR1: Central wavelength: 762 nm. Width: 3.5 nm
IR2: Central wavelength: 763 nm. Width: 8 nm
IR3: Central wavelength: 754 nm. Width: 3 nm
IR4: Central wavelength: 772 nm. Width: 3 nm

alongside a reference in the text:

L78: Comparing MATS images to those from a comparable instrument can help to identify this light and characterise its behaviour. The IR channels of the MATS limb imager are summarised in Table 1.

Regarding the two additional (UV) channels, a short note has been added:

L61: The main channels consist of a narrow spectral window (IR1) and a wider window (IR2). These windows are optimised so that the temperature of the emitting gas can be derived, taking advantage of the wavelength distribution of the emitting $O_2$ gas that depends on its temperature. The instrument further monitors two UV spectral ranges. These channels are named UV1 and UV2, and are designed to target light scattered from noctilucent clouds. The treatment of these channels is outside the scope of this study.

☐ **Section 3.3. line 158. "…40 to 60 km (Figures 4a and 4c).". I would say that Figure 4 displays the selected parameters from 45 km (not from 40 km.)**

This is indeed true. The range on the vertical axis goes from 45 km and not 40 km. The text has been changed accordingly:

L157: For both twilight and daylight conditions, there is good agreement between OSIRIS and SASKTRAN in the intermediate range of 45 to 60 km (Figures 4a and 4c).

☐ **Figure 4. The correspondence between Figure 4b and 4a is clear. The difference between OSIRIS measurement and SASKTRAN model is negligible for altitudes up to nearly 70 km in both Figures. However, the correspondence between Figures 4d and 4c is not so clear. The curves in Figure 4c merge at lower altitudes, whereas the differences in Figure 4d increases at lower altitudes, namely the difference for albedo=0. Is it just because of the logarithmic scale on the horizontal axis, or is it something wrong here? Some explaining comments and/or another way of displaying would be useful.**

We agree that this is confusing. Due to the logarithmic scales in Fig. 4c the differences are hard to see before the difference is computed (Fig. 4d), but as they cover several orders of magnitude a logarithmic scale is required. We have previously experimented with displaying the ratio between OSIRIS and SASKTRAN simulations, but we found that the absolute values and the differences were the most illustrative. As the SASKTRAN simulations are performed in two extremes, albedo 1 and albedo 0, the increasing difference with altitude for the albedo 0 case is

expected, as the OSIRIS measurement is certainly performed in brighter conditions. The 'real' difference is expected to be somewhere between the two displayed cases. We emphasise this in the updated manuscript:

L157: or both twilight and daylight conditions, there is good agreement between OSIRIS and SASKTRAN in the intermediate range of 45 to 60 km (Figures 4a and 4c). This agreement is typically within a few percent, both during twilight and during daylight. In conditions with surface illumination, the level of agreement in the lower altitude range depends largely on the albedo used in the SASKTRAN simulation. Although the exact albedo conditions for the OSIRIS measurement are unknown, it is evident that the measured radiance is larger than that of the dark SASKTRAN case. The actual deviation is expected to be somewhere between the two extremes. At higher altitudes, the signal is dominated by stray light, and the OSIRIS measurements deviate substantially from SASKTRAN.

☐ **Section 4, line 176, "433 MATS dayglow images". There is 1333 in Table 1. What is correct?**

This is an error that has been addressed. The number of dayglow images should be the same as in the table. Double checking these numbers the correct value should actually be 1133 and not 1333. This has been updated in the table and in the text:

L175: Due to the high sampling rate of MATS, approximately 25-65 MATS images are taken within the selection criteria for each OSIRIS measurement, totalling 1133 MATS dayglow images.

☐ **Section 5, line 220, "Figures 7a and 7b were made on 11 January 2023". There is 15 January above Figure 7b.**

This is a mistake - should be 7d. The manuscript has been updated:

L219: Two illustrative examples, from different nightglow conditions, are shown in Figure 7. Figures 7a and 7d were made on 11 January 2023 when the emissions were weaker and more variable. The large red area marks the radiance span between the minimum and maximum of all measurements and illustrates the variability along the orbit.

☐ **Figure 7 caption. "The right column is from a conjunction on 9 January 2023 while the right column is from 25 January 2023". There is two times right column in this sentence and there are other dates above the plots. It should be corrected!**

This is also a mistake that has now been updated. The new caption reads:

Examples of nightglow conjunctions obtained during January. The left column is from a conjunction on 11 January 2023 while the right column is from 15 January 2023. Blue lines illustrate limb radiance measured by OSIRIS. Limb radiances as observed in the left, middle, and right portions of the MATS images are depicted by red, orange, and green lines, respectively. Red shading highlights the range between observed minima and maxima. Inlets display the signal ratios.

☐ **Section 5.1, line 236, "relative difference for both channels" Should not be ratio? Specify!**

We do mean the relative difference. In the previous figures, the inlets showed ratios, but for the statistical analysis the relative difference is more illustrative. This is also highlighted above the inlets as well as in the caption. We have clarified this in the start of Section 5.1.

**Minor comments**

☐ **line 127, of 750 to 775 nm -> from 750 to 775 nm**

Corrected.